# Fabrication of Eco-Friendly Betanin Hybrid Materials Based on Palygorskite and Halloysite

**DOI:** 10.3390/ma13204649

**Published:** 2020-10-18

**Authors:** Shue Li, Bin Mu, Xiaowen Wang, Yuru Kang, Aiqin Wang

**Affiliations:** 1Key Laboratory of Clay Mineral Applied Research of Gansu Province, Center of Eco-Materials and Green Chemistry, Lanzhou Institute of Chemical Physics, Chinese Academy of Sciences, Lanzhou 730000, China; seli17@licp.cas.cn (S.L.); wangxw@licp.cas.cn (X.W.); yurukang@licp.cas.cn (Y.K.); 2Center of Materials Science and Optoelectronics Engineering, University of Chinese Academy of Sciences, Beijing 100049, China; 3Center of Xuyi Palygorskite Applied Technology, Lanzhou Institute of Chemical Physics, Chinese Academy of Sciences, Xuyi 211700, China

**Keywords:** palygorskite, halloysite, betanin, hybrid materials, stability

## Abstract

Eco-friendly betanin/clay minerals hybrid materials with good stability were synthesized by combining with adsorption, grinding, and heating treatment using natural betanin extracted from beetroot and natural 2:1 type palygorskite or 1:1 type halloysite. After incorporation of clay minerals, the thermal stability and solvent resistance of natural betanin were obviously enhanced. Due to the difference in the structure of palygorskite and halloysite, betanin was mainly adsorbed on the outer surface of palygorskite or halloysite through hydrogen-bond interaction, but also part of them also entered into the lumen of Hal via electrostatic interaction. Compared with palygorskite, hybrid materials prepared with halloysite exhibited the better color performance, heating stability and solvent resistance due to the high loading content of betanin and shielding effect of lumen of halloysite.

## 1. Introduction

There has been a long history in the applications of natural plant pigments as colorants for use in the fiber, pottery, paints, and murals, even dating back to prehistoric times [1,2,3,4,5,6]. Several centuries ago, the Mayans in ancient Mesoamerica invented the well-known Maya blue with bright color and excellent stability, which was commonly used in pottery and mural paintings during the late pre-Spanish period [7,8,9]. With the development of the characterization techniques and deeper cognition about interaction mechanisms of Maya blue, much progress has been made in the preparation of synthetic substitutes for natural pigments. Although man-made pigments bring great commercial benefits, they will produce persistent organic pollutants or carcinogens in the production process. 

In recent years, the dye industry has been forced to gradually ban the production of potentially toxic dyes or pigments [10,11], and natural plant pigments have become the focus once again due to the abundance in nature, low toxicity, good biocompatibility and biodegradability, and no oncogenic hazard [4,12,13]. Betalains are water-soluble natural plant pigment containing nitrogen and responsible for the deep yellow or red color of beetroot [14,15,16,17]. Betanin is the main ingredient in betalain, which provides vivid colors for food products as an eco-friendly and safe additive, such as ice cream, yogurt, and fudge, enhancing the visual effects and promoting consumption [16,18,19,20]. However, it has never been found in the same plant as anthocyanin in the plant kingdom. Due to the antioxidative, anti-inflammatory, and anticarcinogenic activities, natural betanin pigment has significant curative effect on the chronic diseases including inflammation, diabetes, cancer, and neurological health [19,21,22]. Increasingly the importance of natural plant pigments toward human health and the environment is being realized, but it is still a challenge to resolve the innate instability and sensitivity against environmental factors (e.g., light, oxygen, pH and metal ions), and fast degradation, which can result in a loss of colors and properties of these natural pigments [15,20,23,24,25,26]. Amjadi et al. studied the inclusion of this bioactive compound in liposomal nanocarriers to improve its digestive stability, antioxidant activity and oral bioavailability [20,27]. However, liposomes have low physical-chemical resistibility and easily degraded at the acidic or high-temperature circumstance [28]. 

Clay minerals have high specific surface area, cation exchange capacity and excellent adsorption properties [29,30,31,32]. In addition, different clay minerals have different morphologies, structures, chemical compositions, and properties [33,34,35]. Therefore, the eco-friendly natural clay minerals have become the focus of the relevant researches on the loading of natural pigments in the last few years, and the interaction between clay minerals and natural pigments is explored and guide to improve the properties of natural pigments. Among them, palygorskite (Pal) is a naturally rod-like hydrated magnesium aluminum phyllosilicate clay mineral with a 2:1 ribbon-layer structure composed of two continuous silica-oxygen tetrahedron sheets and one discontinuous metal-oxygen octahedron sheet [36,37,38,39]. One of the most classic applications of Pal is to synthesize Maya blue pigments as an inorganic host of the stabilizing natural indigo dyes. By contrast, halloysite (Hal) is a natural hollow tubular clay mineral with the external diameter of 20–60 nm, inner lumen larger than 10 nm, and the relatively low surface hydroxyl group density [40,41,42,43]. As a dioctahedral 1:1 clay mineral, Hal is composed of the one tetrahedral SiO_4_ sheet and one octahedral gibbsite Al(OH)_3_ sheet [44,45]. Interestingly, the surface charge of Hal is the result of the interaction between the negatively charged outer layer (siloxane groups) and the positively charged inner surface (aluminol groups) in the range of pH 2–8 [46,47]. At present, our group has successfully synthesized a series of acid/base reversible allochroic anthocyanin/clay minerals hybrid pigments based on anthocyanin natural molecules and clay minerals (Pal, Hal, sepiolite, kaolinite, and montmorillonite) [35,39]. However, the studies on the natural anionic pigments and clay minerals are rarely reported.

Based on above background, the hybrid materials with the expected stability were obtained by loading betanin using Pal and Hal by adsorption, grinding and heating treatment. The environmental stability and possible stability mechanism between clay minerals and betanin molecules were systematically compared and studied in this paper.

## 2. Experimental 

### 2.1. Materials

Pal was obtained from Guanshan Mine, Anhui Province, China, and the main chemical compositions comprised of Al_2_O_3_ (8.28%), Na_2_O (2.15%), CaO (1.28%), MgO (12.29%), SiO_2_ (58.05%), K_2_O (0.92%), and Fe_2_O_3_ (5.04%). Hal was provided by Zhengzhou Jinyangguang Ceramics Co., Ltd., Zhengzhou, China, and the main chemical compositions included Al_2_O_3_ (29.49%), Na_2_O (0.07%), CaO (0.08%), MgO (0.39%), SiO_2_ (42.56%), K_2_O (0.85%), and Fe_2_O_3_ (1.37%). Hal was ground and treated with 4% HCl (wt.%), and then filtered by passing through a 200-mesh sieve and dried. Betanin (red beet extract diluted with dextrin) was provided by Tokyo chemical industry Co., Ltd., Tokyo, Japan. 

### 2.2. Preparation of Betanin/Clay Mineral Hybrid Materials 

At first, 0.75 g of betanin was completely dispersed into 40 mL of distilled water with a pH of 2.0 adjusted by HCl. Then, 1 g of Pal or Hal were slowly added to the above mixture and magnetically stirred for 2 h, respectively, followed by oscillation with 200 rpm at 30 °C for 24 h in a constant temperature shaker (THZ-98A, INESA, Shanghai, China) to reach the adsorption equilibrium. Then, the products were separated by centrifugation at 4000 rpm for 15 min. Subsequently, the wet precipitation was washed with 40 mL of distilled water for 5 min and then centrifuged at 4000 rpm for 15 min. The obtained wet betanin/Pal or betanin/Hal samples were first dried at 40 °C for 5 h in a vacuum drying oven, and then ground in a mortar for 30 min, respectively. After that, the samples were treated at 120 °C for 4 h. Finally, the betanin/Pal and betanin/Hal hybrid materials were obtained after screening through a 100 mesh sieve, respectively. It should be noted that the above operations were carried out in dark conditions. 

### 2.3. Stability Tests

The thermal stability of the as-prepared hybrid materials was tested using a STA449F3 simultaneous thermal analyzer (NETZSCH-Gerätebau GmbH, Selb, Germany) in the temperature range from 30 to 550 °C with a heating rate of 10 °C/min under nitrogen atmosphere. In order to further evaluate the thermal stability of the two samples, 0.04 g of betanin/Pal and betanin/Hal were successively placed in an oven at 90, 120, 150, and 180 °C for 60 min to conduct heat resistance test. The thermal stability of the samples was evaluated by comparing the colorimetric values before and after treatment at different temperatures.

The stability differences of the betanin/Pal and betanin/Hal hybrid materials were also evaluated by immersing in distilled water, acidic, ethanol and basic solutions, respectively. Typically, 0.04 g of samples were dispersed into 20 mL of distilled water, 0.1 M HCl, and 0.1 M NaOH, respectively, and then vibrated at 120 rpm and 25 °C for 24 h in a constant temperature shaker. After centrifugation at 4000 rpm for 15 min, the wet precipitation was completely dried in a vacuum drying oven at 40 °C, followed by the colorimetric values of the dried samples before and after treatment in the above three solutions were measured to study the chemical stability. 

### 2.4. Characterizations

The Fourier transform infrared (FTIR) spectra of samples were recorded on a Nicolet NEXUS FTIR spectrometer (Nicolet iS50, Thermo Scientific, Bartlesville, OK, USA) using KBr pellets in wavenumber range of 400–4000 cm^−1^. The X-ray diffraction patterns (XRD) were record on an X’pert PRO diffractometer using the Cu-Kα radiation in the range of 2*θ* = 3–80 at 40 kV and 40 mA at a scanning rate of 2°/min. The morphologies were taken using the field emission transmission electron microscopy (TEM, Tecnai G 2 F20 S-TWIN TMP, Hillsboro, OR, USA) after the sample was dispersed ultrasonically in anhydrous ethanol and dropped onto a micro grid. The surface area and pore volume of samples were measured using the Accelerated Surface Area and Porosimetry System (Micromeritics, ASAP2020, Atlanta, GA, USA) at −196 °C with N_2_ as an adsorbate. The zeta potentials were measured on a Malvern Zetasizer Nano system (ZEN3600, Malvern, UK) with a 633 nm He-Ne laser irradiated, in which 0.05 g of sample was dispersed into 10 mL of deionized water and sonicated for 20 min before measurement of zeta potential. The surface composition and the chemical state of the samples were investigated by X-ray photoelectron spectroscopy (XPS, ESCALAB 250Xi, ThermoFisher Scientific, Waltham, MA, USA), and all the binding energies were referenced to the C1s peak at 284.8 eV of the surface adventitious carbon. Survey scans were obtained in the 0–1400 eV range, and the high-resolution scanning was recorded for the C1s, N1s, O1s, Si2p, Mg1s and Al2p regions. Thermal gravimetric analysis (TGA) was obtained on a STA449F3 simultaneous thermal analyzer (NETZSCH-Gerätebau GmbH, Wittelsbacherstrasse, Berlin, Germany) at a heating rate of 10 °C min^−1^ under a N_2_ atmosphere. The colorimetric values and reflectance spectra of all samples were calculate on a Color-Eye automatic differential colorimeter (X-Rite, Ci 7800, Pantone Inc., Carlstadt, NJ, USA) according to the Commission International de l’Eclairage (CIE) 1976 *L*a*b** colorimetric method, in which *L** was the color lightness ranging from black (0) to white (100), while *a** and *b** represented the hue with positive and negative values mean red to green and yellow to blue, respectively. The colorimetric values of each sample were tested parallel for three times. The chemical composition was collected from E3 X-ray fluorescence spectrometer (PANalytical, Almelo, The Netherlands). The surface elemental compositions of hybrid materials were determined using a Kevex energy dispersive spectrometer (EDS).

## 3. Results and Discussion

### 3.1. Preparation and Characterization of the Hybrid Materials

The preparation process of betanin/clay mineral hybrid materials are shown in Scheme 1, and the main preparation processes includes adsorption, grinding and heating treatment. Typically, the first step is to load the betanin molecules onto Pal or Hal through a simple adsorption process [39,48]. Subsequently, the grinding and heating treatment are applied to promote the physical and chemical interaction between betanin and clay minerals [39,48]. Among them, the application of heating treatment helps to remove the water molecules partially located in the clay minerals, resulted in facilitating the fixation of betanin molecules on clay minerals and improving their stability [1,8,9,35,39]. The natural Pal and Hal are white and light yellow (Appendix A), respectively, and hybrid materials present pink color after incorporation of betanin, which indicates that betanin has been successfully loaded on the clay minerals [13,35,39]. 

TEM micrographs of betanin/Pal and betanin/Hal hybrid materials exhibit that the typical morphologies of clay minerals is remained after the introduction of betanin molecules (Figure 1a,b) [39,42,48,49,50]. Among them, betanin/Pal hybrid materials present the rod-like morphology, while betanin/Hal hybrid materials exhibit the hollow tubular morphology with the external diameter of 20–60 nm and inner lumen larger than 10 nm. It indicates that the preparation process has no obvious effect on the morphologies of the involved clay minerals.

EDS analysis is used to reveal the surface chemical compositions of betanin/Pal and betanin/Hal samples (Appendix A), and their atomic percent with different elements is presented in Figure 1c,d, respectively. The O, Si, Mg, Al, Fe, and Ca elements are derived from Pal and Hal [34,51,52]. Furthermore, the presence of C and N demonstrates the successful loading of betanin on Pal and Hal. Among them, the detectable amounts of C and N on the Pal surface are 18.01% and 10.89%, respectively, which are higher than that of betanin/Hal (9.82% and 7.92%), indicating that the amount of betanin loading on the Pal surface is higher than that on the external surface of Hal. 

The pore structural parameters of clay minerals and the corresponding hybrid materials are listed in Table 1. Compared with raw clay minerals, the specific surface areas (*S_BET_*) of betanin/Pal sample decreases significantly from 185.85 m^2^/g to 129.19 m^2^/g while that of betanin/Hal hybrid materials decreases by only 9.81 m^2^/g after the incorporation of betanin, indicating that betanin molecules are mainly loaded on the external surface of clay minerals [39,48]. In addition, the total pore volume (*V_total_*) of betanin/Hal sample also reduces to 0.0312 cm^3^/g, whereas the *V_total_* values of betanin/Pal hybrid materials have no significant change. This fact suggests that some betanin molecules might be embedded in the lumen of Hal, resulting in the reduction of the *V_total_* values of Hal. 

As shown in Figure 2, the zeta potentials of betanin at lower pH is −26.43 mV, which is consistent with previous report [53]. After adding betanin, the zeta-potential value of Pal increases slightly from −20.60 mV to −17.80 mV, which is consistent with the results of the stabilization of betacyanins by food grade anionic polysaccharides [53]. Interestingly, the surface charge of Hal is the result of the interaction between the negatively charged external surface (SiO_4_) and the positiviely charged inner surface (AlO_6_) in the range of pH 2–8 [13,42,46]. Therefore, the anion pigment may enter into the lumen of Hal during the preparation process, and the electrostatic interactions occurs within the Hal nanotubes, resulting in a reduction in the positive charge of Hal, and thus the zeta potentials of the hybrid material decreases significantly from −21.83 mV to −32.93 mV. It is also possible that the negatively charged outer surface electrostatically interacts with the positively charged fragments of betanin due to its zwitterionic property. Thus, the carboxyl groups are facing outward, which leads to decreasing in the zeta potential.

As shown in Figure 3a, the XRD patterns of Pal and Hal present typical characteristic diffraction peaks accompanied by a small amount of impurities such as mica and dolomite [48,54]. Pal shows three strong reflection peaks in 2*θ* = 8.42° (*d* = 10.49 Å), 13.79° (*d* = 16.42 Å) and 19.88° (*d* = 4.46 Å), corresponding to (110), (200) and (040) crystal planes [55]. Hal has typical diffraction peaks at 2*θ* = 12.04° (*d* = 7.34 Å) and 24.72° (*d* = 3.60 Å) corresponding to the (001) and (002) planes [56,57]. Compared with the XRD patterns of clay minerals and corresponding hybrid materials, it is found that the XRD patterns of hybrid materials are similar to that of the corresponding clay minerals, no emergence of a new diffraction peak and no obvious layer spacing changes of Pal and Hal in (110) and (001), respectively. 

The FTIR spectra of clay minerals, hybrid materials and natural betanin molecules are shown in Figure 3b. In case of clay minerals, the characteristic absorption bands are presented in Figure 3b. The absorption bands around 3700–3500 cm^−1^ and 3400–3300 cm^−1^ are mainly due to the stretching vibration of the O-H group of the structural hydroxyl group and water molecule [57,58]. In FTIR spectra of Pal, the absorption band at about 1600 cm^−1^ corresponds to the antisymmetric stretching vibration of zeolite water and adsorbed water [48,59,60]. The absorption band of 791~1198 cm^−1^ is the anti-symmetric stretching vibration peak of Si-O bond [59,61]. The bands at 508 and 497 cm^−1^ are assigned to the tetrahedral Si–O deformation and bending vibration, respectively [62]. In FTIR spectrum of Hal, 1637 and 911 cm^−1^ represent the bending vibrations of H_2_O and the deformation of –OH [54,57], while 1097 and 1032 cm^−1^ is attributed to the bending vibration peaks of Si-O and Si-O-Si, respectively [54,56]. In the case of betanin, the absorption band at 3400 cm^−1^ is attributed to the stretching vibration of –OH, the absorption band at 2926 cm^−1^ belongs to the stretching vibration of C–H, and two vibration peaks at 1638 cm^−1^ and 1422 cm^−1^ are ascribed to the –COOH group and the aromatic –CH group, respectively [63]. The symmetric stretching vibration of C–O–C appears at 1024 cm^−1^ [64], while the absorption peak at 707 cm^−1^ confirms the existence of N–H [65]. In the FTIR spectrum of hybrid materials, the absorption peaks of the stretching vibration peaks of O-H bonds in Pal and Hal shift from 3415 cm^−1^ to 3399 cm^−1^ and from 3459 cm^−1^ to 3481 cm^−1^, respectively, which may be due to the hydrogen bond interaction between betanin and clay minerals [27].

The XPS characterization is performed to analyze the existence of surface elements and their chemical states of betanin/Pal and betanin/Hal hybrid materials. As shown in Figure 4a, the signals of Si2p, Al2p, O1s, and C1s are obviously observed from the XPS survey spectra of betanin/Pal and betanin/Hal samples, and Mg1s is only found from the XPS spectra of Pal and betanin/Pal, which is related to the difference in the XRF chemical compositions between Pal and Hal. The N atoms in the chemical composition of the two hybrid materials are hardly detected (Figure 4b and Appendix A), which may be due to the fact that the N element only accounts for 5% of the relative molecular weight of the betanin molecule. Furthermore, the C1s peaks for betanin/Pal and betanin/Hal samples presented at 284.85 eV and 284.83 eV, respectively. In addition, the C1s spectra of the betanin/Pal and betanin/Hal samples consists of the C–C(C–H) and sp2–C, and the corresponding binding energies are located at about 285.20 eV and 284.65 eV, 285.02 eV and 284.66 eV, respectively (Figure 4c). Therefore, it indicates that hybrid materials are successfully prepared based on betanin and Pal or Hal.

In the case of the high-resolution spectra of Si2p of Pal, the binding energies the surface silanol groups and Si–O bonds increase from 103.15 eV to 103.29 eV, and 102.48 eV to 102.72 eV after incorporation of betanin (Figure 4d) [34,66,67], and this change suggests that there is an interaction between the betanin molecules and the surface silanol groups and Si-O of Pal. Meanwhile, the Mg1s binding energy of Pal increases from 1304.20 eV to 1304.23 eV with the introduction of betanin (Figure 4e), while two peaks of Al2p at 74.95 (Al–OH) and 74.53 eV (Al–O) also shifts to 75.42 eV and 74.61 eV, respectively (Figure 4f). It indicates that Mg–O–Si, Al–OH and Al–O–Si of Pal may interact with betanin. It is worth noting that the similar phenomenon is also found from Si2p and Al2p of Hal before and after incorporation of betanin (Figure 4g,h), it reveals that betanin also interact with the surface silanol groups and the Al–OH of the lumen of Hal. Furthermore, the decrease in the intensity of Mg1s and Al2p of of Pal may be due to the leaching of some magnesium and aluminum ions from Pal under acidic conditions, which can be confirmed in Appendix A [66]. 

### 3.2. Thermal and Chemical Stability of the Hybrid Materials

The DSC-TGA curves of clay minerals, hybrid materials and betanin are shown in Figure 5. As depicted in Figure 5a, clay minerals, hybrid materials and betanin have obvious Endothermic peaks in the DSC curves. As can be seen from Figure 5b, natural betanin begins to degrade at room temperature and rapidly lost the weight at about 200 °C, and it presents a maximum loss at 350 °C suggesting that the degradation of natural pigments basically completes. At 550 °C, the residual mass of betanin is only 25.57%, indicating that natural pigments have poor high temperature resistance. In case of raw Pal, the degradation process occurs in the three stages of 30–550 °C, which are related to the removal of the adsorbed water, zeolitic H_2_O, the bound OH_2_, and part of structural OH, respectively [39,48,60,62]. As for raw Hal, it shows an initial slight mass loss from room temperature, which is attributed to the evaporation of the physically adsorbed water and the interlayer H_2_O. In addition, the weight loss from 400 °C to 550 °C is ascribed to the removal of hydroxyl groups of Hal [13,68,69]. Obviously, the hybrid material exhibits different degradation behavior to the corresponding clay minerals, which is due to the addition of betanin. The betanin in betanin/Pal and betanin/Hal sample exhibits the obviously low weight loss rate of 1.35% and 0.30% at 200 °C, respectively. It is worth noting that the final mass loss of betanin in betanin/Hal (1.31%) is significantly lower than that of in betanin/Pal (6.07%). Therefore, the results demonstrate that betanin/Hal presents much better thermal stability than betanin/Pal at higher temperatures. It is possibly due to the stronger physical and chemical interactions between Hal and betanin molecules, and the fact that the small molecules partly enters into the lumen of Hal, resulting in more superior shielding effect.

The chromaticity values of betanin and hybrid materials after being successively treated at 90, 120, 150, and 180 °C for 60 min are applied to further evaluate the thermal stability of samples. As shown in Figure 6a, the changes of *L**, *a*,* and *b** of betanin at lower heating temperature are very obvious, indicating that the thermal stability of betaine is poor. After incorporation of clay minerals, the chromaticity values of the hybrid material changes little within 90 °C (Figure 6b,c). As the temperature continues to rise, the *L** and *b** values of betanin/Pal increase gradually, while the *a** values decreases gradually. Compared with betanin and betanin/Pal, betanin/Hal has a smaller change in chromaticity value exhibiting the better thermal stability, as evidenced by digital images of samples at different heating temperatures (Appendix A). It might be due to the fact that betanin is not only adsorbed on the external surface of Hal, but also confined in the lumen [54].

In addition, the solvent resistance of hybrid material is also evaluated. The colorimetric parameters of hybrid materials before and after being immersed into distilled water, 0.1 M hydrochloric acid and sodium hydroxide for 24 h are listed in Table 2. As shown in Table 2, the color parameters of betanin/Pal samples before and after immersion in the three solutions are significantly different. After being treated with hydrochloric acid and sodium hydroxide, the *a** value decreases significantly to 1.89 and 1.48, while *b** increases from 5.99 to 13.00 and 9.98. The CIE parameters of betanin/Hal hybrid materials change less than those of betanin/Pal samples after being treated with different solutions. Combined with the images of the supernatants after being immersed in the above solutions (Appendix A), it can be concluded that the introduction of Hal is more helpful to improve the stability of natural pigments in distilled water, acid and alkaline solutions.

It is clear that the color of betanin/Hal after being treated by distilled water, 0.1 M HCl and 0.1 M NaOH for 24 h is much closer to the red area than that of hybrid materials under the same treatment condition (Figure 7a). Furthermore, the characteristic absorption bands in the range of 450–600 nm representing red are attributed to the existence of betanin in the diffuse reflectance spectra (Figure 7b,c). The intensity of absorption bands of betanin/Hal samples is larger than that of betanin/Pal samples after being treated using different solvents for 24 h (Figure 7b,c). This further confirms that Hal has a better shielding effect on natural pigments toward the external environments. 

## 4. Conclusions

The betanin/clay mineral hybrid materials were successfully prepared using the natural betanin molecules and Pal or Hal. It was found that the thermal and chemical stability of the betanin molecule was also obviously improved after incorporation of clay minerals, and the betanin/Hal hybrid materials presented the better color performance (*L** = 62.64, *a** = 15.43, *b** = 6.18) than betanin/Pal samples (*L** = 64.94, *a** = 14.96, *b** = 5.99). Due to the differences in the structure of clay minerals, the betanin molecules were mainly loaded on the surface of Pal or Hal, but part of betanin also entered into the lumen of Hal, and thus betanin/Hal hybrid materials exhibits the better color property and stability against the external environmental factors. Furthermore, the as-prepared stable hybrid materials may be a potential candidate as eco-friendly antibacterial agent to be applied in relevant fields combining with the inherent antibacterial and nontoxic components.

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
