# Peer review of "Fabrication of Eco-Friendly Betanin Hybrid Materials Based on Palygorskite and Halloysite"

_materials, 2020, doi:10.3390/ma13204649_

Round 1

Reviewer 1 Report

Eco-friendly betanin hybrid materials based on palygorskite and halloysite

Shue Li, Bin Mua, Xiaowen Wanga, Yuru Kang and Aiqin Wang

In this article, Eco-friendly betanin/clay minerals hybrid materials with good stability were synthesized combining natural betanin molecules extracted from beetroot with 2:1 type palygorskite (Pal) and 1:1 type halloysite (Hal), respectively.

The manuscript is interesting, original and in the scoop of the journal.

My comments :

  1. Hal is a commonly used abbreviation for halogen. Recommend using a another abbreviation for halloysite.
  2. Lines 186-188: “After adding betanin, the zeta-potential value of Pal increases slightly from -20.60 mV to -17.80 mV, which was consistent with the results of the stabilization of betacyanins by food grade anionic polysaccharides. “ In this case, the authors talk about an increase in the value of the ζ-potential and conclude about an increase in stability. However, in fact, the stability of the colloidal system is characterized by the ζ-potential modulus. Moreover, the system will be highly stable if the ζ-potential magnitude is greater than 30mV. In this case, the magnitude of the ζ-potential decreases with the addition of betanin.
  3. What does R2+ and R3+ mean in Scheme 1?
  4. In accordance with the DLS data, the authors believe that a significant decrease in potential (from -21.83 mV to -32.93 mV) is due to the participation of the positively charged inner surface in electrostatic interaction with the anionic pigment. However, it is a little bit strange to call betaine an anionic pigment. It is a zwitterionic compound, i.e. simultaneously contains positively and negatively charged groups. And in this case, the explanation can be completely opposite: the negatively charged outer surface electrostatically interacts with the positively charged fragments of betaine (N +). Carboxyl groups are facing outward and this causes a decrease in the zeta potential. I think that the explanation given in the article is not entirely correct.

Author Response

Question (1):Hal is a commonly used abbreviation for halogen. Recommend using a another abbreviation for halloysite.

Reply:Thanks for your comments and suggestion. In fact, Hal is also commonly used as an abbreviation for halloysite in clay minerals [1-3].

[1] Peng, H.X.; Zhang, D, Liu,X.H.; Tang, W; Wan, H.; Xiong, H.; Ma, R.Z.Facile synthesis and characterization of core-shell structured Ag3PO4@Hal nanocomposites for enhanced photocatalytic properties. Appl. Clay Sci. 2017, 141, 132-137.

[2] Zhuang, G.Z.; Jaber, M.; Rodrigues, F.; Rigaud, B.; Walter, P.; Zhang, Z.P. A new durable pigment with hydrophobic surface based on natural nanotubes and indigo: Interactions and stability. J. Colloid Interf. Sci.2019, 552, 204-217.

[3] Zhang, Y.M.; Li, Y.Q.; Zhang, Y.F. Preparation and intercalation structure model of halloysite-stearic acidintercalation compound. Appl. Clay Sci.2020, 187, 105451.

Question (2):Lines 186-188: “After adding betanin, the zeta-potential value of Pal increases slightly from -20.60 mV to -17.80 mV, which was consistent with the results of the stabilization of betacyanins by food grade anionic polysaccharides. “ In this case, the authors talk about an increase in the value of the ζ-potential and conclude about an increase in stability. However, in fact, the stability of the colloidal system is characterized by the ζ-potential modulus. Moreover, the system will be highly stable if the ζ-potential magnitude is greater than 30mV. In this case, the magnitude of the ζ-potential decreases with the addition of betanin.

Reply:Thanks for your comment. Compared with Pal, Hal has the negatively charged outer layer (siloxane groups) and the positively charged inner surface (aluminol groups). Upon incorporation of betanin, betanin molecules could enter into the lumen of Hal, which resulted in a reduction in the positive charge of Hal, and thus the zeta potentials of the hybrid material composed of betanin and Hal obviously decreased. Furthermore, the involved stability in the study referred to the environment stability of natural betanin instead of colloidal stability, and the improvement of the stability of natural betanin to external environments was attributed to the shielding effect of the lumen of Hal.

Question (3):What does R2+ and R3+ mean in Scheme 1?

Reply:Thanks for your comment. R2+ and R3+ represent divalent or trivalent metal ions in the structure of clay minerals, such as Mg2+, Fe2+, Fe3+, and Al3+.

Question (4):In accordance with the DLS data, the authors believe that a significant decrease in potential (from -21.83 mV to -32.93 mV) is due to the participation of the positively charged inner surface in electrostatic interaction with the anionic pigment. However, it is a little bit strange to call betaine an anionic pigment. It is a zwitterionic compound, i.e. simultaneously contains positively and negatively charged groups. And in this case, the explanation can be completely opposite: the negatively charged outer surface electrostatically interacts with the positively charged fragments of betaine (N +). Carboxyl groups are facing outward and this causes a decrease in the zeta potential. I think that the explanation given in the article is not entirely correct.

Reply:Thanks for your suggestion and comment. It has been supplemented in the paper and highlighted in red.

Reviewer 2 Report

The reading of this paper is interesting and there are good points developed in terms of significance of content.

Nevertheless, some important issues must be improved.

  1. Both the title and the abstract are misleading and do not clearly define which is the topic of the paper. The authors just mention "eco-friendly materials" and speak about thermal stability and color performance. Conversely, they present many materials characterization results: the quality of the paper could be improved if those results were discussed. Also, they do not properly discuss why their materials are eco-friendly.
  2. The experimental section should be improved to provide all the details. In particular, XPS experimental parameters are not defined making the discussion on the results not supported.
  3. Lines 229-256: This discussion must be improved. The Authors state that N1s spectra are reported in Figure 4, but this spectral region is depicted in Figure S2. Also, they are wrong in stating that C binding energies are detected in Figure S2. The N1s binding energy can be hardly evidenced by N1s spectra because no peak is there, and in case the reported BE should be compared with N1s spectrum from free betanin. The absence of surface nitrogen is in contrast with what is stated in lines 167-168. I know that EDS and XPS have different sampling depth, but no explanation is given for nitrogen absence on the surface, despite the claimed surface adsortpion. For the same reason, N1s spectrum deserves to be in the main paper. 
    All the spectra in Figure 4 should be revised for the curve fitting procedure and the background subtraction.
    The Authors report BE values with a precision that is not supported  either from associated errors or spectral resolution. Also, the chemical environment they claim must be carefully checked.
    The Authors comment on "peak areas", but the atomic percentages should be calculated and provided in order to compare Hal and Pal amount of functional groups.
  4. Conclusion section: as for abstract, this section is not coherent with the paper results.

Author Response

Question (1):Both the title and the abstract are misleading and do not clearly define which is the topic of the paper. The authors just mention "eco-friendly materials" and speak about thermal stability and color performance. Conversely, they present many materials characterization results: the quality of the paper could be improved if those results were discussed. Also, they do not properly discuss why their materials are eco-friendly.

Reply:Thanks for your comments. The title of this manuscript has been revised into “Fabrication of eco-friendly betanin hybrid materials based on palygorskite and halloysite”, and the topic of this manuscript is focused on the preparation of hybrid materials using natural betanin pigment and natural clay minerals of palygorskite or halloysite. The materials used in this paper are derived from natural plant pigments and clay minerals, both of them are safe and non-toxic. In addition, the preparation process is simple and pollution-free. Therefore, the obtained samples are environmentally friendly.

Question (2):The experimental section should be improved to provide all the details. In particular, XPS experimental parameters are not defined making the discussion on the results not supported.

Reply:Thanks for your comments. It has been supplemented in the characterization and highlighted in red.

Question (3):Lines 229-256: This discussion must be improved. The Authors state that N1s spectra are reported in Figure 4, but this spectral region is depicted in Figure S2. Also, they are wrong in stating that C binding energies are detected in Figure S2. The N1s binding energy can be hardly evidenced by N1s spectra because no peak is there, and in case the reported BE should be compared with N1s spectrum from free betanin. The absence of surface nitrogen is in contrast with what is stated in lines 167-168. I know that EDS and XPS have different sampling depth, but no explanation is given for nitrogen absence on the surface, despite the claimed surface adsortpion. For the same reason, N1s spectrum deserves to be in the main paper.

All the spectra in Figure 4 should be revised for the curve fitting procedure and the background subtraction.

The Authors report BE values with a precision that is not supported either from associated errors or spectral resolution. Also, the chemical environment they claim must be carefully checked.

The Authors comment on "peak areas", but the atomic percentages should be calculated and provided in order to compare Hal and Pal amount of functional groups.

Reply:Thanks for your comments. The errors of color parameters of hybrid materials before and after being treated by chemical solvent has been revised in Table 2. The N element accounts for only 5% of the relative molecular weight of the betanin molecule, so the nitrogen content is very low. But it can still be detected in the chemical composition of the samples, and the atomic percentages has been listed in Table S2.

Question (4):Conclusion section: as for abstract, this section is not coherent with the paper results.

Reply:Thanks for your comment. In this paper, the hybrid materials with good stability were synthesized combining natural betanin molecules with 2:1 type Pal and 1:1 type Hal, respectively. The environmental stability and possible interaction mechanism between clay minerals and betanin molecules were compared and studied. The structural characterizations confirmed that betanin was mainly adsorbed on the outer surface of Pal or Hal through hydrogen-bond interaction, and part of them also were confined into the lumen of Hal via electrostatic interaction. Therefore, Hal preferred to improve the color properties, heating stability and solvent resistance of natural betanin due to its structural features compared with Pal.

Round 2

Reviewer 2 Report

The Authors have fulfilled most of the requests, but still the XPS data presentation and explanation is non adequate.

Author Response

Question (1): The Authors have fulfilled most of the requests, but still the XPS data presentation and explanation is non adequate.

Reply: Thanks for your comments and suggestion. It has been supplemented in the paper and highlighted in red.

Round 3

Reviewer 2 Report

The Authors still do not provide proper answers/corrections to the XPS part, disregarding the issue already raised in the first reviwe report. Please find in the following the questions that still do not have any answer.

1. All the spectra in Figure 4 should be revised for the curve fitting procedure and the background subtraction.

The Authors did not accomplish nor answer to this issue. In particular, fig 4h is completely unacceptable from the background subtraction point of view.

As for the curve fitting, the Authors apply this procedure also to Si2p and Al2p regions disregarding that those signals are partially unresolved doublet: they should either use the doublets or preliminarly perform spin-orbit splitting subtraction.

2. The Authors report BE values with a precision that is not supported either from associated errors or spectral resolution. Also, the chemical environment they claim must be carefully checked.

The Authors report BE values with two decimal points, without expressing the associated errors (neither as step point acquisition nor as std deviation). I can hardly believe to those numbers if resolution parameter is not expressed. Also, they claim chemical environment attribution and BE changes relying on BE values that are not significantly different. Again, the ultimate resolution of their XPS experiments should be taken into proper considertaion. 

3. The Authors comment on "peak areas", but the atomic percentages should be calculated and provided in order to compare Hal and Pal amount of functional groups.

They still refer to peak areas.

Moreover, in Table S2 they report N quantification, but from the N1s signal reported in the paper (Fig. 4b) it can hardly detected the nitrogen presence. For sure the atomi percentages reported in the table cannot be derived from those signals.

Based on the aforementioned considerations, the whole XPS part should be revised and properly interpreted and commented.

The paper cannot be accepted in the present from.

Author Response

Question (1): All the spectra in Figure 4 should be revised for the curve fitting procedure and the background subtraction. The Authors did not accomplish nor answer to this issue. In particular, fig 4h is completely unacceptable from the background subtraction point of view. As for the curve fitting, the Authors apply this procedure also to Si2p and Al2p regions disregarding that those signals are partially unresolved doublet: they should either use the doublets or preliminarly perform spin-orbit splitting subtraction.

Response: Thanks for your comments and suggestion, and Fig. 4 has been revised according to the reviewer’s suggestion, while the curve fitting of Si2p and Al2p in Fig. 4 also has been re-fitted, and the results are consistent with the following literatures [1,2].

References:

[1]Lu, Y.S.; Dong, W.K.; Wang, W.B.; Ding, J.J.; Wang, Q.; Hui, A.P.; et al. Optimal synthesis of environment-friendly iron red pigment from natural nanostructured clay minerals. Nanomaterials (Basel), 2018, 8, 925.

[2] Wang X.W.; Mu, B.; Zhang, A.J.; An, X.C.; Wang, A.Q. Effects ofdifferent pH regulators on the color properties of attapulgite/BiVO4 hybrid pigment. Powder Technology, 2019, 343, 68-78.

Question (2): The Authors report BE values with a precision that is not supported either from associated errors or spectral resolution. Also, the chemical environment they claim must be carefully checked. The Authors report BE values with two decimal points, without expressing the associated errors (neither as step point acquisition nor as std deviation). I can hardly believe to those numbers if resolution parameter is not expressed. Also, they claim chemical environment attribution and BE changes relying on BE values that are not significantly different. Again, the ultimate resolution of their XPS experiments should be taken into proper consideration.

Response: Thanks for your comments and suggestion. The relevant resolution parameters are provided as follows: True control system performance: 5×10-10 mbar, best spatial resolution of monochromator: ≤ 20 microns, best energy resolution of monochromator: ≤ 0.45 eV, and monochromator (large area) energy resolution: 400000 cps (FWHM≤0.50 eV).

Question (3): The Authors comment on "peak areas", but the atomic percentages should be calculated and provided in order to compare Hal and Pal amount of functional groups.
They still refer to peak areas. Moreover, in Table S2 they report N quantification, but from the N1s signal reported in the paper (Fig. 4b) it can be hardly detected the nitrogen presence. For sure the atom percentages reported in the table cannot be derived from those signals.
Reply: Thanks for your comments and suggestion. The XPS analysis has been rewritten from the direction of the binding energy rather than peak areas and can demonstrate certain interactions between betanin and clay minerals [3]. The issue of N quantification has been reviewed in the paper and highlighted in red.

[3]Dong, W.K.; Lu, Y.S.; Wang, W.B.; Zhang, M.M.; Jing, Y.M.; Wang, A.Q. A sustainable approach to fabricate new 1D and 2D nanomaterials from natural abundant palygorskite clay for antibacterial and adsorption. Chemical Engineering Journal, 2020, 382, 122984.

Your efforts in the review process of this manuscript are greatly appreciated. If you have any question about the article, please don’t hesitate to contact me.